# Programmable DNA looping using engineered bivalent dCas9 complexes

Nan Hao [1], Keith E. Shearwin[1] & Ian B. Dodd[1]

DNA looping is a ubiquitous and critical feature of gene regulation. Although DNA looping can be efficiently detected, tools to readily manipulate DNA looping are limited. Here we develop CRISPR-based DNA looping reagents for creation of programmable DNA loops. Cleavage-defective Cas9 proteins of different specificity are linked by heterodimerization or translational fusion to create bivalent complexes able to link two separate DNA regions. After model-directed optimization, the reagents are validated using a quantitative DNA looping assay in *E. coli*. Looping efficiency is ~15% for a 4.7 kb loop, but is significantly improved by loop multiplexing with additional guides. Bivalent dCas9 complexes are also used to activate endogenous *norVW* genes by rewiring chromosomal DNA to bring distal enhancer elements to the gene promoters. Such reagents should allow manipulation of DNA looping in a variety of cell types, aiding understanding of endogenous loops and enabling creation of new regulatory connections.

[1] Department of Molecular and Cellular Biology, School of Biological Sciences, The University of Adelaide, Adelaide, SA 5005, Australia. Correspondence and requests for materials should be addressed to N.H. (email: nan.hao@adelaide.edu.au)

Transcription is frequently regulated positively or negatively by the interaction of proteins bound to different sites on the same DNA molecule. These regulatory interactions can occur at long range (>1 kb), forming large DNA loops[1–3]. In multicellular organisms, many promoters are regulated by distal enhancer or silencer elements, sometimes separated by megabase distances[4, 5]. Similarly, in bacteria, the activity of σ54-dependent promoters can be activated by bEBPs (bacterial enhancer-binding proteins) bound many kb away from the promoter[2]. Though poorly understood, the efficiency and specificity of these interactions appears to be controlled by other DNA looping elements. In eukaryotes, dedicated tethering elements are used to stimulate specific enhancer–promoter contacts[6], while insulator elements interact with each other to sequester the enhancer and promoter into separate DNA loops[7]. Genome-wide chromatin capture studies have revealed a vast number of highly specific long-range DNA–DNA interactions, supporting a ubiquitous role of DNA looping in gene regulation[8]. However, the function of most of these DNA loops, and how these highly specific patterns of DNA looping are achieved and modulated, remains unclear.

Engineering of artificial DNA loops provides a means to manipulate endogenous DNA loops to better understand their function and to modulate gene expression, potentially with therapeutic applications, such as forming new enhancer–promoter connections to circumvent genetic deficiencies or inhibiting dysfunctional enhancer–promoter contacts. For example, fusion of the Ldb1 protein that binds to proteins at β-globin promoters to an engineered zinc finger DNA-binding domain able to bind to the β-globin locus control region (LCR) was able to redirect the LCR to activate an alternative promoter[9]. However, the design of zinc finger proteins able to target specific sequences can be laborious, and in the above case the looping reagent required a pre-existing protein–protein interaction. An alternative approach, insertion of binding sites for heterologous DNA-looping proteins, requires alteration of the genome sequence. Thus, a heterologous reagent that can be readily programmed to connect any two endogenous DNA segments would greatly facilitate DNA loop engineering.

Fusions of various protein domains to cleavage-defective Cas9 proteins (dCas9s) from bacterial CRISPR (clustered regularly interspaced short palindromic repeat) systems have been widely used to target specific functionalities to specific genomic DNA sequences, directed by the sequence of the associated single guide RNA (sgRNA)[10]. A bivalent CRISPR reagent that contains two orthogonal dCas9 moieties should thus be capable of programmed binding to two DNA segments at the same time by using two different sgRNAs. Here we design and construct such a reagent and show that it can drive de novo DNA loops in *E. coli* cells. We show that this DNA looping reagent is highly programmable and also multiplexable, readily allowing simultaneous linking of multiple targets to promote efficient DNA looping over a ~12 kb distance.

## Results

**Modeling DNA looping by a bivalent dCas9 complex.** Our first approach was to construct dCas9 heterodimers (Fig. 1a). The use of two orthogonal dCas9s allows the DNA-binding specificity of each dCas9 component of the heterodimer to be independently programmed by its specific cognate sgRNA. A homodimeric design, though simpler, has the disadvantage that half of the homodimers would be charged with sgRNAs of the same sequence specificity and would inhibit DNA looping. We chose the well-studied *Streptococcus pyogenes* (*Sp*) and *Streptococcus thermophilus* CRISPR1 (*St*) dCas9s[11].

In order to optimize the efficiency of looping, we developed a simple statistical mechanical model for DNA looping by a bivalent dCas9 (Fig. 1a, b). The model indicates that the fractional looping $F_{loop}$, that is the fraction of time that the DNA loop is formed, is critically dependent on five parameters: (1) the dimerization of the *Sp* and *St* dCas9 monomers, given by the dimer dissociation constant $K_{dim}$ (nM); (2) the binding affinity of each dCas9–sgRNA complex for its DNA site, given by a dissociation constant, $K_{DNA}$ (nM); (3) the fractional saturation of each of the dCas9 proteins with its cognate sgRNA, $\theta$; (4) the total concentrations of each dCas9, $[C_{tot}]$ (nM); and (5) the DNA loopability factor $J$ (nM), the effective relative concentration of one target DNA site relative to the other. At distances beyond 250 bp in vivo, $J$ is primarily dependent on the length of the DNA loop and decreases as the loop gets larger, due to the entropic cost of holding two distant DNA sites together[12–14].

In our model, we made the simplifying assumption that $K_{DNA}$, $\theta$, and $[C_{tot}]$ are the same for both *Sp* and *St* dCas9s (Fig. 1b–e). Looping efficiency depends on all five parameters, such that the exact response of looping to changes in any one parameter depends on the values set for the other four parameters. Nevertheless, with the exception of dCas9 concentration, which has a clear optimum, all the other parameters show a monotonic effect on DNA looping (Fig. 1c).

As expected, DNA looping improves with stronger dimerization between dCas9 monomers, such that the majority of dCas9 complexes exist in a looping competent dimer form rather than looping-incompetent monomers (Fig. 1c). To this end, we fused strong heterodimerizing leucine zippers[15] to the C-termini of *Sp* and *St* dCas9 (dCas9_Zip). These synthetic leucine zipper tags contain four pairs of electrostatically attractive salt bridges, designed to destabilize homodimer interactions and maximize inter-helical interactions. The strong dimerization constant for this pair, $K_{dim} = 1.3 \times 10^{-2}$ nM[15], should be close to optimal for DNA looping (Fig. 1c).

Since only sgRNA-bound dCas9 can bind the DNA target[16, 17], looping is maximized when the dCas9s are fully saturated with sgRNA (Fig. 1c). To attempt to ensure saturation, we expressed the *Sp* and *St* dCas9_Zip proteins from chromosomally integrated single-copy constructs, while the sgRNAs were expressed from a multicopy plasmid (p15a ori, 15–20 copies per cell) under the control of a strong synthetic promoter.

Our model also predicts that DNA looping increases with higher affinity interactions between the dCas9–sgRNA complex and its target DNA (i.e., low $K_{DNA}$; Fig. 1c). There is some uncertainty about the $K_{DNA}$ of *Sp* dCas9 in vivo. The *Sp* dCas9–sgRNA complex was shown to interact strongly with a perfectly matched DNA site in vitro, with a $K_{DNA}$ ~1.3 nM by bio-layer interferometry[18]. However, a substantially weaker $K_{DNA}$ of 68.28 nM was obtained using microscale thermophoresis[19]. Furthermore, a $K_{DNA}$ of 105 nM was obtained by fitting in vivo repression data[20]. We used this in vivo estimate for the plots in Fig. 1c–e.

The model further predicts that there exists an optimal dCas9 concentration for looping (Fig. 1c), as seen for looping by Lac repressor[13, 21]. If the concentration of dCas9 is too low, then its fractional occupation of the target DNA is sub-optimal for looping. Conversely, if the dCas9 concentration is too high, then both target sites will tend to be separately occupied by a dCas9 dimer (Fig. 1b, species 12–15), rather than having a single dCas9 dimer that bridges the two DNA sites. In the case where dCas9 is fully dimerized and saturated with guide RNA, looping is optimal when the dCas9 concentration $[C_{tot}]$ is equal to its $K_{DNA}$. It follows that the lower the absolute values of $K_{DNA}$ (and $[C_{tot}]$) are, the greater the efficiency of looping (Fig. 1d). To keep the dCas9_Zips in the concentration range where efficient looping is

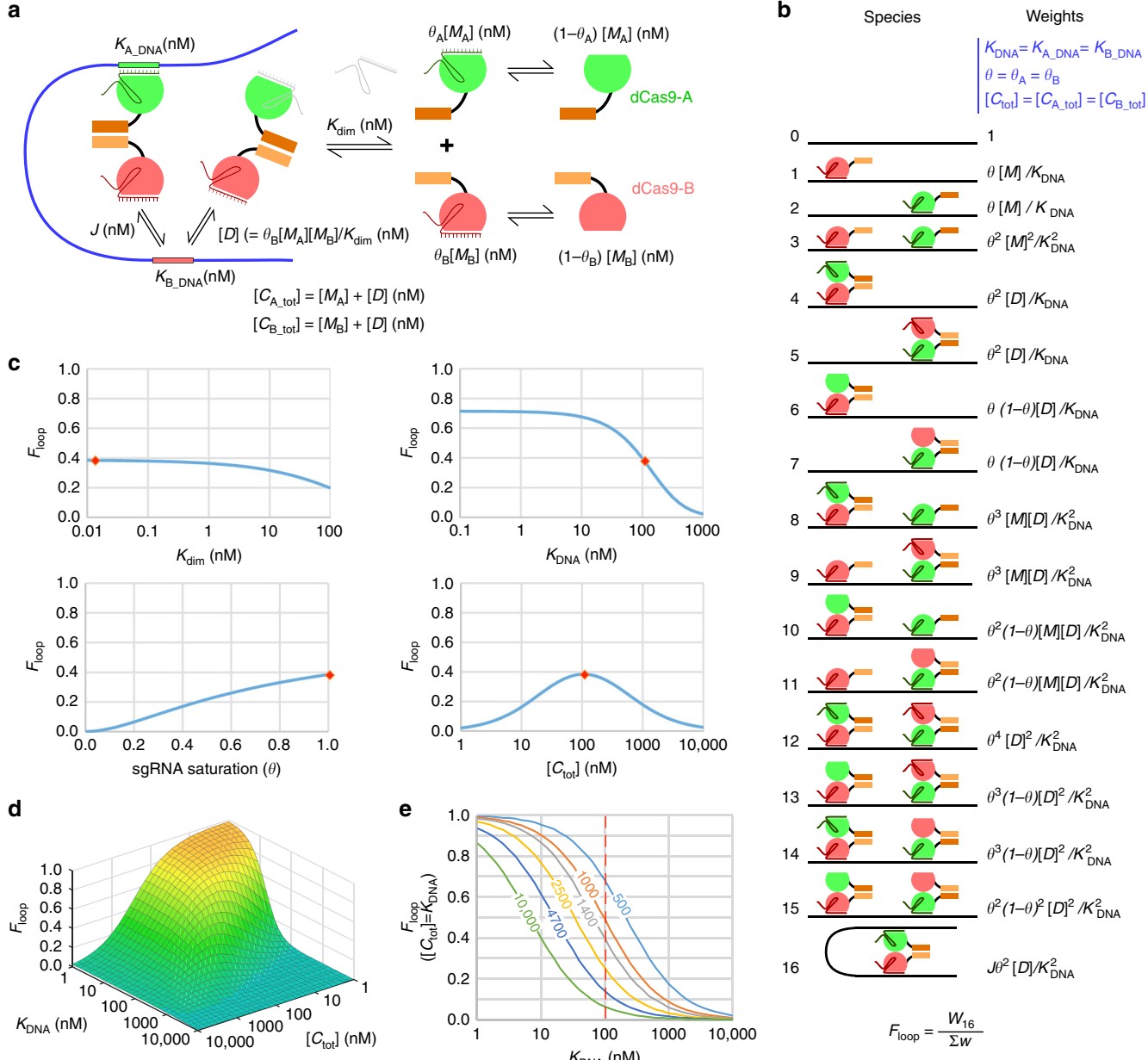

**Fig. 1** Modeling dCas9-mediated DNA looping. **a** Model parameters. $K_{dim}$ dimerization constant, $K_{DNA}$ dissociation constant of dCas9 to DNA, $J$ the effective concentration of one target DNA site relative to the other, $[M]$ concentration of dCas9 monomer, $[D]$ concentration of dCas9 dimer, $[C_{tot}]$ total concentration of one dCas9, and $\theta$ the fractional saturation of dCas9 with its cognate sgRNA. Subscripts A and B denote two orthogonal dCas9s. **b** Statistical mechanical model of DNA looping by heterodimerizing dCas9s, assuming the same parameters for each dCas9. **c** The effect of varying $K_{dim}$, $K_{DNA}$, $\theta$, and $[C_{tot}]$ on $F_{loop}$. In each of the plots, only one parameter is changed while the others are kept constant: $K_{dim} = 1.3 \times 10^{-2}$ nM[15], $K_{DNA} = 105$ nM[20], $\theta = 1$, $[C_{tot}] = 1.05 \times 10^2$ nM, $J = 266$ nM at $d = 1400$ bp separation ($J = 1.08 \times 10^6 \times d^{-1.15}$)[13]. For the case where $[C_{A\_tot}] = [C_{B\_tot}]$, $[M]$ for each dCas9 was calculated as $((\sqrt{(1 + 4[C_{tot}]/K_{dim})})-1)/(2/K_{dim})$ and $[D]$ as $[C_{tot}] - [M]$. **d** At constant $K_{DNA}$, $\theta$, and $J$, $F_{loop}$ increases when $K_{DNA}$ decreases. At any given $K_{DNA}$, $F_{loop}$ is highest when $[C_{tot}] = K_{DNA}$ (assuming complete dimerization). **e** At optimal dCas9 concentration ($[C_{tot}] = K_{DNA}$), $F_{loop}$ is inversely related to $K_{DNA}$. At any given $K_{DNA}$, the maximal $F_{loop}$ decreases as distance separation between two DNA sites increases (i.e., decreasing $J$). At $[C_{tot}] = K_{DNA} = 1.05 \times 10^2$ nM (marked by the red line), the maximal $F_{loop}$ is estimated to be 67, 48, 39, 24, 14, and 6% at 0.5, 0.8, 1.4, 2.5, 4.7, and 10 kb separation between the two looping sites

possible, we used three different promoters with roughly a 25:3:1 ratio in their respective promoter strengths[22] to produce a range of dCas9_Zip concentrations.

Finally, the model shows that the maximal achievable looping efficiency is also dictated by $J$. The plots of Fig. 1c assume $J = 266$ nM, which is the expected value for a 1400 bp loop based on the empirical relationship between $J$ and loop length obtained from measurements of LacI and λ CI looping in *E. coli* cells[13]. The plots

of Fig. 1e show the expected maximal DNA looping for a range of loop lengths for bivalent dCas9 of different DNA-binding affinities. For a given $K_{DNA}$, DNA looping is favored with a smaller separation (higher $J$ values). For a given loop length (a fixed $J$), lower $K_{DNA}$ (and $[C_{tot}]$) values improve looping. This is because looping is a competition for binding to the free target site between a dCas9 dimer bound at the other target site, at

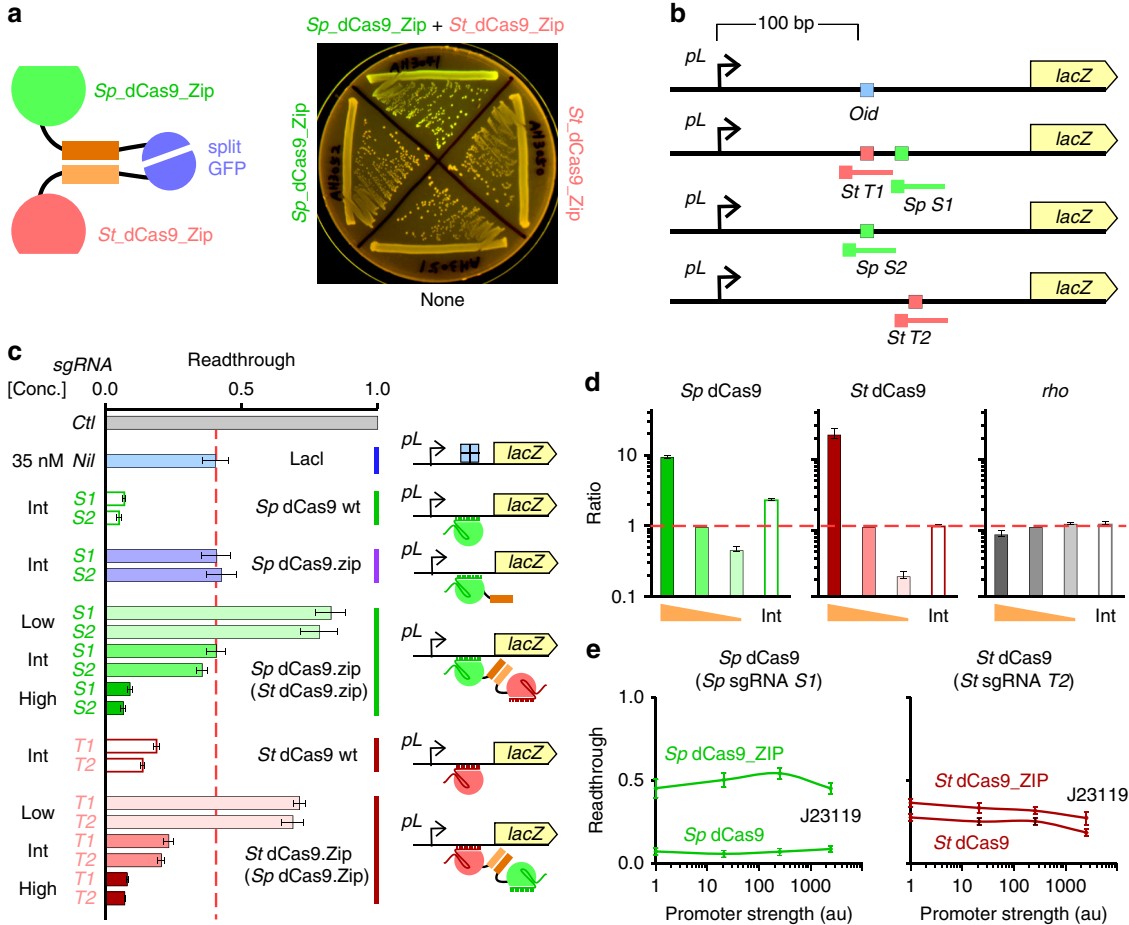

**Fig. 2** Characterization of the system components. **a** Split-GFP assay confirming heterodimerization of *Sp* and *St* dCas9_Zip in vivo. **b** Schematic of the chromosomally integrated roadblocking reporter constructs (see also Supplementary Fig. 1). The relative positions of sgRNA-targeting sites are marked with lines. The PAM proximal end is boxed. **c** Roadblocking readthrough across *lacOid* with 35 nM LacI or dCas9 target sites with various concentrations of wt (unshaded) or Zip fusion (shaded) dCas9s. Both *Sp* and *St* dCas9s were expressed from chromosomally integrated expression modules under the control of low, intermediate (Int), or high strength promoters. Readthrough values are calculated as steady state LacZ units obtained in the presence of roadblock proteins (LacI or dCas9) divided by the activity in their absence. Thus, readthrough = 1 when there is no roadblocking, and readthrough = 0 when there is complete roadblocking. Data are mean ± 95% confidence intervals (*n* = 9). **d** qRT-PCR showing the relative expression level of the wt (unshaded) and Zip fusion (shaded) dCas9 under the control of low, intermediate, and high strength promoters. The *E. coli gyrA* gene was used as a reference gene for data normalization and the *rho* gene was used as an internal control. Data are mean ± standard error (*n* = 3) and are expressed as a ratio relative to the intermediate expression level of dCas9_Zip. **e** Roadblock readthrough at different sgRNA expression levels. The relative expression levels are 1:21:256: ~2500, with the strongest promoter being J23119 (Biobricks, http://parts.igem.org/Part:BBa_J23119)

concentration *J*, and a dCas9 dimer in solution, at concentration [*D*], with looping favored when *J* > [*D*] (Fig. 1a).

**Testing the system components.** To assess the ability of dCas9_Zips to heterodimerize in vivo, split-GFP (green fluorescent protein) assays[23] were used. The *Sp* and *St*_dCas9_Zips were fused with the C- or N-terminal halves of GFP, respectively (Supplementary Fig. 2). Co-expression of plasmid-borne *Sp* dCas9_Zip_GFP_C and *St* dCas9_Zip_GFP_N, but not any of the individual fusion proteins alone, led to the reconstitution of a functional GFP protein (Fig. 2a), suggesting that the *Sp* and *St* dCas9_Zips are capable of heterodimerization in vivo.

To confirm DNA binding of the dCas9_Zips in vivo, transcriptional roadblocking assays[24] were used, where dCas9_-Zips were directed to target the 5′ UTR of the *lacZ* reporter gene (Fig. 2b). The principle of this assay is that a DNA-bound complex at a transcribed region can stall elongating RNA polymerase (RNAP), leading to transcription termination and reduced expression of the reporter. Strong roadblocking by dCas9

proteins has been demonstrated in *E. coli*, particularly when the sgRNA targets the non-template strand[25–27].

Both *Sp* and *St* dCas9_Zips expressed from the intermediate strength promoter blocked *lacZ* transcription to a level comparable to or below that seen with ~35 nM of Lac repressor and its high-affinity *lacOid* operator (in vivo $K_{DNA}$ ~0.17 nM[28]) (Fig. 2c). There was no detectable difference in roadblocking by an *Sp* dCas9_Zip protein alone or *Sp* dCas9_Zip in the presence of its heterodimerizing partner protein *St* dCas9_Zip. However, roadblocking by *Sp* dCas9_Zip was ~sixfold weaker than that of wt *Sp* dCas9 when expressed from the same intermediate strength promoter, potentially due to its twofold reduced expression level (Fig. 2d). In contrast, roadblocking by *St* dCas9 and its Zip fusion (+heterodimerizing partner) was similar when expressed from the intermediate strength promoter. As expected, roadblocking by the dCas9_Zips was relatively insensitive to the two different but fully complementary spacer sequences tested, however, roadblocking was strongly affected by the expression levels of dCas9_Zips. At the high expression level, both *Sp* and *St* dCas9_Zips were capable of blocking *lacZ* transcription by ~90%, while at the low

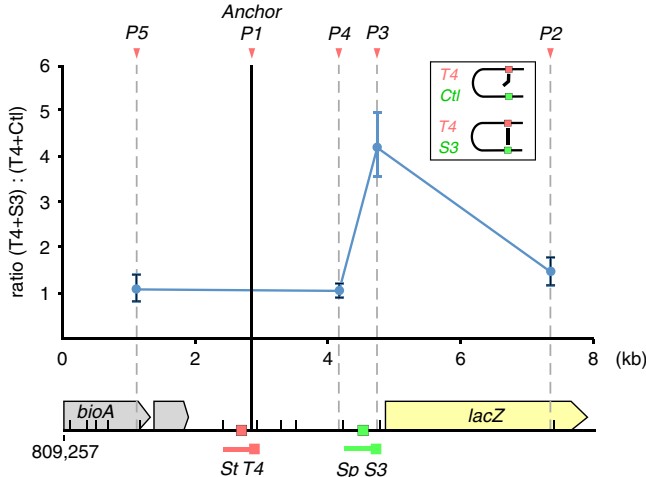

**Fig. 3** Assaying dCas9-mediated looping via chromosome conformation capture (3C). Relative contact frequencies between the anchor fragment (solid black line) and four of the BsaWI restriction digest fragments (dotted gray lines) were assessed by qPCR, and plotted as the ratio between cells expressing a pair of specific sgRNAs (T4 + S3) and cells expressing only one specific sgRNA (T4 + Ctl). The sgRNA T4 is the guide RNA that localizes dCas9_Zip heterodimer to the anchor fragment. Data are mean ± standard deviation (n = 3). The DNA region of interest is shown below the plot, with the dCas9 target sites highlighted, and the BsaWI restriction sites indicated by short black bars. Orange arrowheads above the plot indicate the qPCR primer sites

expression level, *Sp* and *St* dCas9_Zips gave only 5–15% and ~30% roadblocking, respectively (Fig. 2c).

In addition, transcriptional roadblocking assays also confirmed that both *Sp* and *St* dCas9_Zips were saturated with their cognate sgRNAs, as roadblocking was not decreased even when the promoter expressing the sgRNAs was weakened ~2500-fold (Fig. 2e and Supplementary Fig. 2).

**The dual-dCas9 complex is capable of looping DNA in vivo.** To test whether our engineered bivalent dCas9_Zip complex was capable of looping DNA, chromatin configuration capture (3C) assays were performed. The dCas9_Zip heterodimers were expressed from an intermediate strength promoter, and paired with either T4 + S3 or T4 + Ctl sgRNAs (Fig. 3). In contrast to T4 + Ctl control, where the dCas9_Zip heterodimers were only localized to the chromosomal DNA by a single site, targeting dCas9_Zip to both T4 and S3 sites significantly increased the contact frequency between these two sites, 1.7 kb apart, demonstrating the formation of a dCas9_Zip-mediated DNA loop between these two sites. However, while the 3C assay is commonly used to demonstrate looping between two regions of DNA[9, 29], it does not give a quantitative measure of looping efficiency.

To answer this question, we turned to a second in vivo looping assay, based on our previous demonstration that a DNA loop nested inside another DNA loop can assist the outer loop's formation[14, 30]. Loop assistance occurs because looping between the two internal sites reduces the effective distance between the two outside sites, making them more likely to interact with each other (Fig. 4a).

We designed pairs of *Sp* and *St* sgRNAs that targeted two DNA sites about 1.4 kb apart from each other within a 2.1 kb LacI looping reporter (Fig. 4b). The formation of a LacI-mediated loop between the distal *lacOid*-binding site and the promoter–proximal *lacO2* site can be detected by its effect on

promoter repression[13, 28, 30, 31]. In the absence of an upstream operator, LacI binds weakly to the *lacO2* operator, leading to partial repression of the *lacZ* promoter at the low [LacI] used. In the presence of the strong distal *lacOid* operator, LacI bound at that site can interact by DNA looping to simultaneously occupy the *lacO2* site, increasing total *lacO2* occupancy and promoter repression (Supplementary Fig. 3A, B). By comparing promoter activity in the presence or absence of the distal site, it is possible to determine the absolute fraction of LacI looping $F_{Lac}$[13]. Loop assistance due to looping between the internal sites increases $F_{Lac}$ and is revealed by increased LacI-mediated repression of the reporter[14, 30].

As shown in Fig. 4b, a consistent decreased expression of the reporter was observed for all couplings of the bivalent dCas9_Zip and specific *Sp* and *St* sgRNAs targeting two internal DNA sites (T2 + S1, T3 + S1, T2 + S2, and T3 + S2). In contrast, targeted wild-type *Sp* and *St* dCas9 proteins did not affect reporter activity. Expression of the bivalent dCas9_Zip with control sgRNAs, targeting either no internal site or any single internal site, also did not affect the reporter (Fig. 4b). Similarly, a further set of control experiments in which the reporter DNA was altered to remove all but one dCas9 target site (Supplementary Fig. 4), confirmed that two targeted DNA sites were needed for the effect. This dependence on each of the components required for dCas9-mediated looping—the targeted DNA sites, the correct sgRNAs, and the heterodimerizing dCas9-Zips—shows that the decreased reporter activity is due to dCas9 looping.

As expected from our model, the degree of loop assistance changed with different expression levels of the dCas9_Zips. In all cases, the dCas9 loop was most readily formed with the intermediate expression level, enhancing $F_{Lac}$ from 73 ± 3% (95% confidence intervals, n = 9) to 83 ± 2%. A model for loop assistance[30] estimates that this improvement in LacI looping is equivalent to ~40% looping by dCas9 alone (Fig. 4c and Supplementary Fig. 3D), which corresponds well with the 40% maximal looping expected from our dimeric dCas9 model (Fig. 1e). Higher expression levels did not increase loop assistance further, despite showing a stronger roadblocking effect in the roadblock assay (Fig. 2c), suggesting that the dCas9_Zip concentration begins to exceed its optimum (Fig. 1c), which leads to each of the looping sites being occupied by a separate dCas9_Zip dimer and consequent loop breakage. Loop assistance at the lowest expression level of dCas9_Zip heterodimer only slightly increased *lac* loop formation to 77 ± 2% (Fig. 4b), consistent with a low occupation of target DNA at this concentration (Fig. 2c).

The result also showed that at all three dCas9 expression levels, similar loop assistance was observed when the two halves of a dCas9_Zip dimer targeted either the same strand of DNA or opposite strands of the DNA (Fig. 4b, e). This flexibility increases the programmability of the system, as PAM sequences on either DNA strand can be utilized as valid target sites.

To further examine this programmability of looping, we designed two *St* sgRNAs that bind ~150 and 290 bp downstream of the distal *lacOid* operator, and paired each of them with either control *Sp* sgRNA or one of five specific *Sp* sgRNAs that target within a ~300 bp window upstream of the p*lacUV5* promoter (loop sizes for the dCas9 loops ranging from 1400 to 1870 bp). Using the intermediate expression level of the dCas9_Zips, we showed that DNA looping is extremely robust, with all combinations of specific *Sp* and *St* sgRNAs consistently assisting the formation of a 2.1 kb *lac* loop from ~73 to 82% (Fig. 4e).

**DNA looping by an *Sp_St*_dCas9 fusion.** An alternative approach to making a bivalent dCas9 is to simply fuse the two

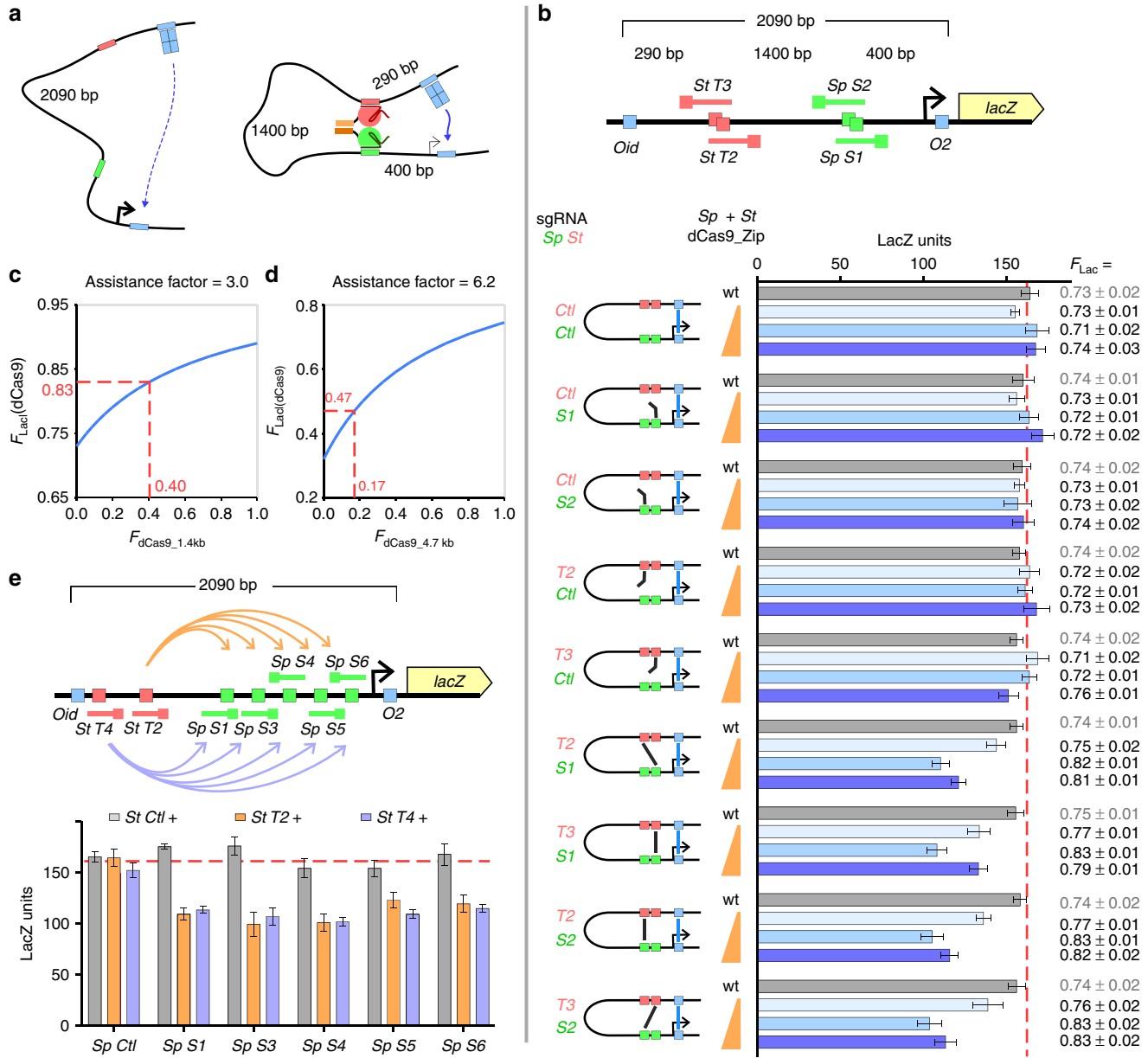

**Fig. 4** Assaying dCas9-mediated looping via loop assistance. **a** Basic principle of the loop assistance assay. **b** Formation of the CRISPR loop assists formation of the LacI loop. $F_{loop}$ is calculated as $[(LacZ_{OD-} - bkgd) - (LacZ_{OD+} - bkgd)]/(LacZ_{OD-} - bkgd)$ (Supplementary Fig. 3A–C). The measured average $LacZ_{OD-}$ values for wt as well as different concentration of $Sp/St$ dCas9_Zip fusions were $578 \pm 17$, $549 \pm 18$, $541 \pm 22$, and $567 \pm 30$, respectively. Data are mean $\pm$ 95% confidence intervals ($n = 9$). A $bkgd$ value = 16 units (unrepressible LacZ background) is estimated[13]. All $F_{loop}$ values are mean $\pm$ standard deviations ($n = 9$). **c, d** The $F$ for the CRISPR loop alone, $F_{dCas9}$, can be calculated from the observed LacI looping in the presence of dCas9, $F_{LacI(dCas9)}$, by applying a loop assistance factor equivalent to the distance shortening due to the formation of the internal loop[30]. **e** The formation of the CRISPR loop is relatively insensitive to the spacer sequences used. Data are mean $\pm$ 95% confidence intervals ($n = 9$)

dCas9 monomers together as a single polypeptide, in effect making the dimerization infinitely strong (Fig. 1c). To test DNA looping by this alternative design, we joined the two proteins by a 15 amino acid flexible linker, and expressed it from a medium copy plasmid (CloDF13 ori, 20–40 copies per cell) under the control of strong, intermediate, and weak promoters. The need for plasmid-based expression and the different expression promoter response compared to the dCas9_Zips probably reflects difficulties in expressing the 2504 amino acid fusion protein. In the loop assistance assay, DNA looping by this fusion dCas9 was able to increase $F$ for a 2.1 kb *lac* loop from $72 \pm 3$ to $82 \pm 2\%$ (Supplementary Fig. 5), similar to that achieved with the $Sp + St$ dCas9_Zip heterodimer. DNA looping by the fusion protein was strongest when expressed from the weak *pro1* promoter, followed by the intermediate *proA* promoter, and was weakest with the strong *proC* promoter (Supplementary Fig. 5), suggesting that the expression level from both intermediate and strong promoters exceeds the optimum for looping (Fig. 1c).

## Enhancing DNA looping via dCas9-mediated loop multiplexing.
We have demonstrated that both $Sp + St$ dCas9_Zip heterodimers and $Sp\_St$ dCas9 fusions were able to form DNA loops within a 2.1 kb LacI loop to assist *lac* loop formation from ~73 to 83% (Fig. 3b and Supplementary Fig. 5). However, loop assistance by the dCas9 complexes was relatively weak, as at this small distance, DNA looping is already quite efficient with LacI

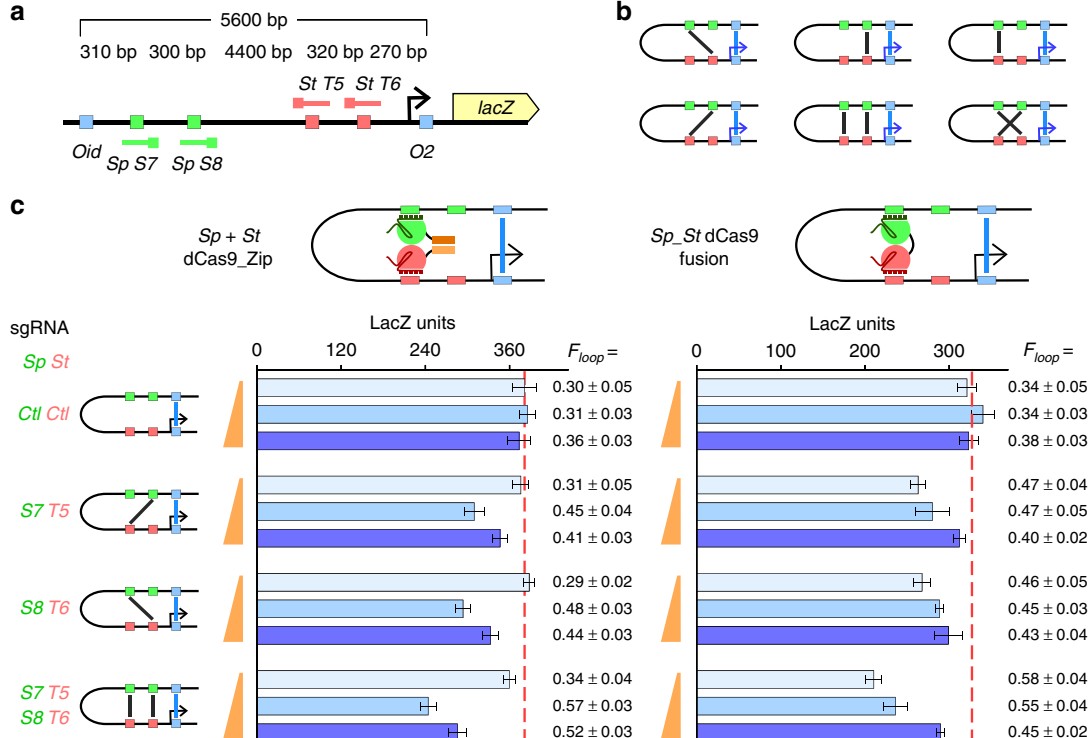

**Fig. 5** Long-range DNA looping can be enhanced via dCas9-mediated loop multiplexing. **a** Schematic representation of the 5.6 kb loop assistance reporter, with the relative positions of the sgRNA-binding sites highlighted. **b** Possible arrangements of CRISPR loops in the presence of four guide RNAs. **c** Single or multiplexed CRISPR loops formed by heterodimerizing dCas9_Zip complexes or the *Sp_St* dCas9 fusion can increase LacI looping efficiency. Data are mean $\pm$ 95% confidence intervals ($n = 9$). $F_{loop}$ values are mean $\pm$ standard deviations ($n = 9$)

alone. Even if the dCas9 loop were to form 100% of the time, the maximal looping expected for the *lac* loop would be 89% (Fig. 4c). To test the effectiveness of our CRISPR looping reagents over a larger distance, we employed an additional chromosomally integrated *lacZ* reporter with a *lacOid* operator 5.6 kb upstream of the *lacO2* operator (Fig. 5a). Thus, looping sizes for sgRNAs S7 + T5, and S8 + T6 are both around 4.7 kb.

As expected, DNA looping by LacI alone becomes much weaker at 5.6 kb, at only ~34 ± 5%, consistent with our previous findings[13]. Loop assistance by a single pair of *Sp* and *St* sgRNAs combined with either *Sp* + *St* dCas9_Zip heterodimer or *Sp_St* dCas9 fusion increased LacI looping (Fig. 5c). Similar to the previous results (Fig. 4b and Supplementary Fig. 5), loop assistance was strongest for the *Sp* + *St* dCas9_Zip heterodimers expressed from a single-copy intermediate strength promoter, and for the *Sp_St* dCas9 fusions expressed by a weak promoter from a medium copy number plasmid. When expressed at their optimal concentrations, both *Sp* + *St* dCas9_Zip heterodimer and *Sp_St* dCas9 fusion improved the LacI looping from ~34 to 47%. Applying a similar analysis to that used in Fig. 4c, we estimate that the bivalent dCas9 is alone capable of ~17% looping at a 4.7 kb distance (Fig. 4d and Supplementary Fig. 3D). This is again consistent with the ~14% maximal looping at this distance predicted from our dimeric dCas9 model (Fig. 1e).

In addition, the ability of the DNA-binding specificity of each dCas9 element to be independently programmed allows for the possibility of increasing overall loop efficiency by creating multiple loops simply by expressing multiple sgRNAs. To test this multiplexibility, we simultaneously expressed two sgRNAs for each dCas9, which should allow the dCas9 looping reagent to form four different single loops and two possible double loops (Fig. 5b). Each of these nested looping species should assist the formation of the outside LacI loop. Indeed, expression of all four

sgRNAs in the 5.6 kb reporter was able to increase *lac* looping ~1.7-fold to ~58% (Fig. 5c), a substantial improvement from the single dCas9-mediated loop assistance.

**The dual-dCas9 loops improves enhancer–promoter contact.** As an independent assay for bivalent dCas9-mediated DNA looping, a classical bacterial promoter–enhancer interaction model[32] was adapted. Here a $\sigma^{54}$-dependent *glnAp2* promoter was used to drive the expression of a chromosomally integrated *lacZ* reporter gene. The activity of *glnAp2* is regulated by the NtrB-NtrC bacterial two-component signal transduction system. The NtrB is both a kinase and a phosphatase, which regulates the phosphorylation status of NtrC. Phosphorylated NtrC is a transcriptional activator that binds to enhancer elements and interacts with $\sigma^{54}$-RNAP bound at the promoter to catalyze its transition to the open complex[33] (Fig. 6a). The kinase/phosphatase activity of NtrB is normally controlled by the carbon and nitrogen status of the cell. A single mutation of NtrB (A129T) significantly reduces the phosphatase activity of NtrB without affecting its kinase activity, resulting in constitutively active NtrC[34]. Here we used recombineering to introduce the A129T mutation into the endogenous NtrB gene (*glnL*) and showed that the expression of *glnAp2* driving the *lacZ* reporter gene is strictly dependent on the presence of the NtrC enhancer element. As expected, as the separation between the enhancer and the promoter is increased, the effect of the enhancer diminishes (Fig. 6b). When placed 11.7 kb away from the promoter, the effect of the enhancer site is only ~2% of that seen when it is placed ~300 bp away.

However, this enhancer-mediated reporter gene activation can be significantly increased when the enhancer site is brought closer to the promoter using either the *Sp* + *St* dCas9_Zip heterodimer or the *Sp_St*_dCas9 fusion (Fig. 6c, d). The effect of bivalent dCas9 is ~twofold at 5.6 kb when a pair of specific sgRNAs were

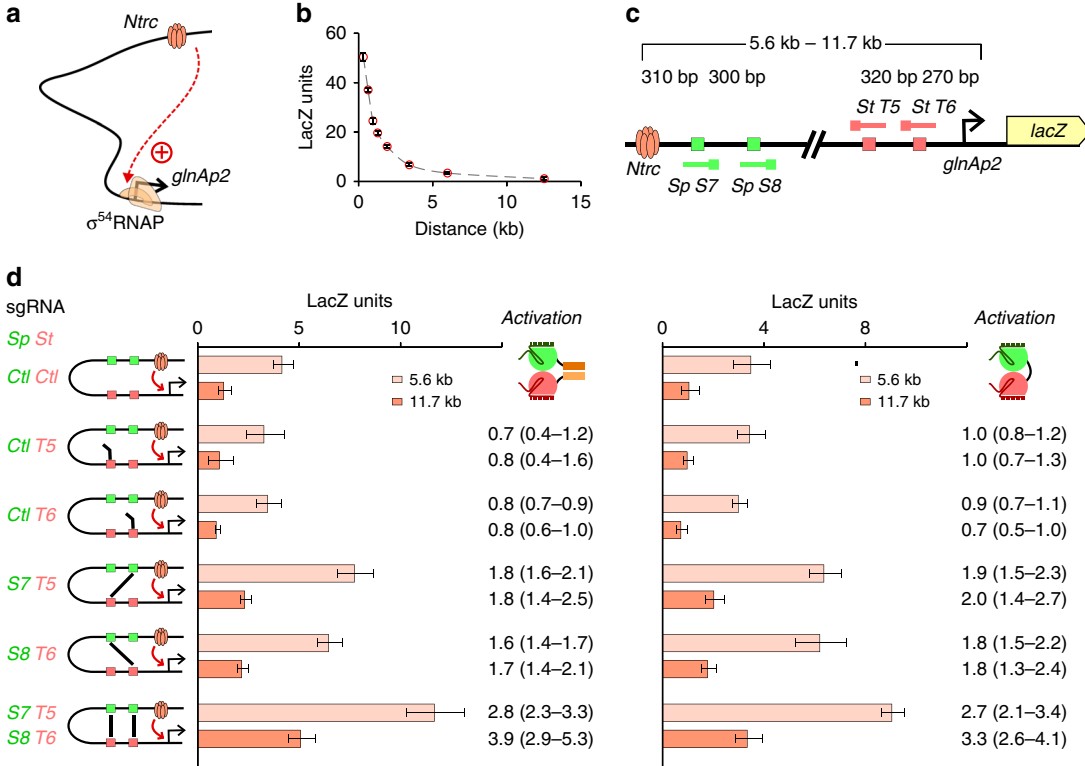

**Fig. 6** dCas9-mediated DNA looping can be used to foster bacterial enhancer–promoter communication. **a** An NtrC hexamer bound at a distal enhancer element can interact with $\sigma^{54}$-RNAP bound at the promoter to active gene transcription. **b** The activation effect of NtrC decreases as the distance between the enhancer and the promoter is increased. Data are mean ± 95% confidence intervals ($n = 9$) with the background activity in the absence of the enhancer element subtracted. **c** Schematic representation of the 5.6 and 11.7 kb looping reporter used for testing dCas9-mediated promoter–enhancer activation. **d** Single or multiplexed CRISPR loops formed by heterodimerizing dCas9_Zip complexes or the *Sp_St* dCas9 fusion increase *glnAp2* transcription by looping out DNA between enhancer and promoter. Data are mean ± 95% confidence intervals ($n = 9$). Numbers indicate fold activation relative to control sgRNAs, with 95% confidence limits in brackets

supplied and this effect increased to ~threefold with dual-dCas9-mediated loop multiplexing using four specific sgRNAs (Fig. 6c, d). Similar results were also obtained at a 11.7 kb separation between the enhancer and promoter sites (Fig. 6c, d). In contrast, when only one specific sgRNA is expressed (together with a control sgRNA), binding of bivalent dCas9 to the promoter–proximal site without making a distal site contact did not lead to increased *lacZ* expression, confirming that the increased *glnAp2* transcription is a result of bivalent dCas9-mediated DNA looping.

To test whether the CRISPR looping reagents could also be used to regulate a $\sigma^{54}$-dependent promoter at its endogenous locus, we inserted an NtrC enhancer element 2 kb upstream of the *norVW* promoter in the NtrB(A129T) strain, and designed a pair of sgRNAs that targeted two DNA sites about 1 kb apart located between the enhancer and the promoter (Fig. 7a). Again, if the bivalent dCas9 complex is capable of looping DNA, it will bring the enhancer site closer to the promoter, and should increase promoter activation (Fig. 7b). The endogenous enhancer for the *norVW* promoter is not expected to be active under the growth conditions used, and insertion of the active NtrC enhancer gave ~10-fold activation of *norVW* transcription (Fig. 4c). More importantly, the formation of the dCas9 loop between the enhancer and promoter further increased the *norV* and *norW* gene expression by ~twofold, suggesting that the bivalent dCas9-based programmable DNA looping reagent is capable of looping DNA in vivo to modulate endogenous gene expression.

## Discussion

In this study, we demonstrated that engineered bivalent dCas9 complexes, consisting of heterodimerizing dCas9s from *S. pyogenes* and *S. thermophilus* or a translational fusion of these two proteins, were capable of driving DNA loop formation in vivo. The efficiency of DNA looping was estimated at ~40 and 17% at 1.4 and 4.7 kb, respectively. DNA looping was further improved by multiplexing (Figs. 4c and 5d). For example, formation of two dCas9-mediated loops, using four sgRNAs, significantly increased a 11.7 kb enhancer–promoter interaction, leading to ~threefold activation of reporter gene expression (Fig. 6d). The improvement in DNA looping with loop multiplexing suggests that higher efficiencies and longer distance DNA looping could be achieved by programming even more target sites.

It may also be possible to improve the efficiency of formation of individual loops by modifying the bivalent dCas9 reagents. Our modeling and data suggest that looping by our dCas9 reagents is unlikely to be limited by a lack of sgRNA charging, poor dimerization, or non-optimal concentration. Rather, the fairly weak $K_{DNA}$ value of 105 nM for dCas9–sgRNA DNA binding in vivo, obtained from an independent estimate for *Sp* dCas9[20] and consistent with our data, is likely to be limiting (Fig. 1d, e). Stronger DNA binding by the dCas9–sgRNA elements, together with lower dCas9 concentrations, should produce large improvements in looping efficiency (Fig. 1d). One potential avenue for improving binding is to explore different dCas9_Zip fusion designs. Another approach would be to test different

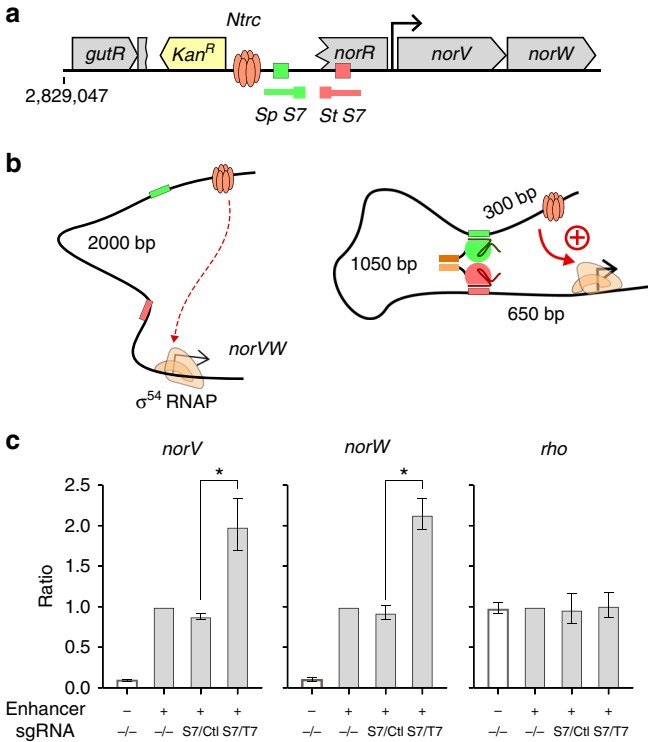

**Fig. 7** dCas9-mediated chromosome rewiring of an endogenous gene. **a** Schematic illustration of the *gutRQ-norRVW* loci of the engineered *E. coli*. Recombineering was used to replace the *gutQ-norR* genes with a module containing NtrC enhancer-binding sites and a kanamycin selection cassette (yellow). Gray boxes represent endogenous genes. **b** The principle of the rewiring experiment. Bivalent dCas9-mediated loop assistance is expected to bring the NtrC sites closer to the $\sigma^{54}$-controlled *norVW* promoter, leading to improved activation. **c** qRT-PCR showing the relative expression levels of the *norV*, *norW*, and *rho* genes in the wild-type (unshaded) or engineered (shaded) *E. coli* strains, either in the presence or absence of specific sgRNAs. Improved activation of the *norV* and *norW* genes (but not the *rho* control gene) is seen only when both specific guides are present. The *E. coli gyrA* gene was used as a reference gene for data normalization. Data are mean ± standard error ($n = 3$) and are expressed as a ratio relative to the engineered strain in the absence of sgRNAs. Asterisk indicates $p < 0.05$ evaluated by a paired one-tailed Student's *t* test on the $\log_{10}$ transformed data

CRISPR-Cas proteins, e.g., *Sa*Cas9, Cpf1, or C2c1[35–37], that may bind DNA more strongly than the *Sp* and *St* dCas9s. A third approach could be to mutate dCas9 to increase DNA-binding affinity. Structural studies of Cas9 in complex with sgRNA and target DNA indicate that the binding energy of the complex includes both sequence-specific and non-specific contributions[38], [39]. Recently, two independent reports showed that the non-specific Cas9–DNA interactions can be reduced by mutating hydrophilic amino acids that stabilize the non-target DNA strand[40] or that contact the phosphate backbone of the target DNA strand[41]. Thus, it may be possible to take the reverse approach and introduce hydrophilic amino acids to strengthen non-specific DNA interactions, and thus overall DNA binding. For example, the *Sp* Cas9 S845K and L847R mutations, which decrease specificity[40], may strengthen DNA binding. An increase in off-target binding may not be problematic for looping applications, since increased specificity is provided by the requirement for two DNA targets.

Given the demonstrated cross platform compatibility of CRISPR-Cas techniques[42–45], we envisage that our CRISPR looping reagents will be readily transferable to other bacterial and eukaryotic systems. The programmability of the CRISPR DNA looping reagents may be readily expanded by substituting either or both the *S. pyogenes* and *S. thermophilus* dCas9_Zips with other dCas9 proteins with different PAM specificities, as long as these proteins are orthogonal with regard to binding guide RNA and exhibit strong DNA binding. Indeed, while our manuscript was under revision, a related paper[29] has demonstrated the fusion of dCas9 from *S. pyogenes* and *S. aureus* with ligand-inducible heterodimerization tags from the plant phytohormone S-(+)-abscisic acid signaling pathway, and showed that dCas-mediated DNA looping can be used to re-establish β-globin gene expression in K562 human erythroleukemia cells by bringing the LCR to the β-globin promoter. Thus, dCas9-mediated looping works in both prokaryotic and eukaryotic settings.

In addition, bivalent dCas9 reagents have potential applications beyond engineering DNA loops. They could be used to bring together separate DNA molecules in order to improve DNA repair or DNA insertion—for example, after cleavage mediated by active Cas. Also, a bivalent Cas9 reagent will have a higher affinity for its DNA targets compared to a monovalent Cas9, as long as the two target sites are close enough to favor simultaneous binding, since dimerization and DNA looping generates cooperative binding. Thus, a dCas9–Cas9 bivalent reagent where the uncleaved and cleaved target sites are near to each other could provide a simple way to improve the efficiency of on-target DNA cleavage without causing increased off-target cleavage, a major issue for CRISPR-based genome editing[46–48]. The specificity of genomic targeting of transcription-regulating domains using dCas9 fusions[10] is also likely to be improved by use of bivalent dCas9–dCas9 reagents targeted to two nearby DNA sites.

## Methods

**Bacterial strains.** EC100D *mcrA Δ(mrr-hsdRMS-mcrBC) φ80dlacZΔM15 ΔlacX74 recA1endA1 araD139 Δ(ara,leu)7697 galU galK λ⁻ rpsL nupG pir⁺ (DHFR)* (Epicentre) was used for propagation of *R6γK ori* (*pir*-dependent) plasmids. Other plasmids were propagated in DH5α *F⁻ Φ80lacZΔM15 Δ(lacZYA-argF) U169 recA1endA1 hsdR17 (rK⁻, mK⁺) phoA supE44 λ⁻ thi-1 gyrA96 relA1*.

E4643 is BW30270 (MG1655 *rph+*) with the endogenous *lacIZYA* locus (EcoCyc MG1655: 360, 527–366,797) removed by recombineering using a 90bp oligonucleotide that targets the lagging strand of the replication fork, with 45bp of homology either side of the deletion site. The successful recombinant was selected as a white colony on an LB agar plate containing X-gal (20 μg mL⁻¹), and the sequence across the deletion junction verified[49]. The parental strain for roadblocking assays, 3C assays, and loop assistance assays was AH4213, a derivative of E4643, with a LacI expression module consisting of the native *PlacI* promoter and *lacI* gene, integrated at the primary bacteriophage 186 *attB* site (Supplementary Fig. 1B).

The parental strain for enhancer activation assays was *E. coli* strain AH5244, a derivative of E4643 containing the *glnL*:A129T mutation (also known as NRII2302) generated by scarless recombineering[50]. Briefly, the wild type *glnL* gene in E4643 was first disrupted by insertion of a *kanR-P_{rhaB}-tse2* module containing a kanamycin resistance gene and a rhamnose inducible *P_{rhaB}* promoter driving the *tse2* toxin gene, and selected on LB agar plate supplemented with 2% glucose and kanamycin (20 μg mL⁻¹). A second round of recombineering was then performed with a 90bp oligonucleotide that harbours *glnL*:A129T mutation, and counter-selected on 1.5% M9 minimal medium (M9MM) agar supplemented with 2 mM MgSO₄, 0.1 mM CaCl₂, and 0.2% rhamnose. The successful recombinant strain was sequence verified.

**DNA construction.** DNA constructions used commercial DNA synthesis (Integrated DNA Technologies, GenScript), restriction enzyme-based cloning, and isothermal Gibson assembly. All sequences are available on request.

The reporter and expression constructs were made using a plasmid integration system developed from the CRIM plasmid series[51]. As in the CRIM system, transiently expressed phage integrase proteins were used to integrate the plasmids into phage attachment sites in the bacterial chromosome. Integration was at λ attB (EcoCyc MG1655 sequence position: 806551), φHK022 attB (position 1055419), φP21 attB (position 1196214), or the primary φ186 attB site (position 2783828).

PCR was used to screen for correct single-copy integrants[52]. Insertions into the bacterial chromosome by recombineering utilized the pSIM6 plasmid[53].

**Reporter constructs.** All roadblocking assay reporters (Supplementary Fig. 1) were derived from pIT3-CL_λpL_lacOid_lacZ*(O2−)[24], an integratable plasmid carrying a λ phage pL promoter (−73 to +90), a lacOid operator (centered at +113 position), and an RNaseIII cleavage site upstream of a lacO2− lacZ reporter gene. The lacZ* gene carries a weakened ribosome-binding site (nucleotides at positions +289, +302, and +304 were mutated from C, T, C to A, G, and G, respectively). To assay the ability of Sp and St dCas9 or dCas9_Zip to bind DNA, the 20 bp lacOid operator was replaced by a synthetic DNA sequence containing protospacers for (1) Sp sgRNA S1 and St sgRNA T1; (2) Sp sgRNA S2; and (3) St sgRNA T2 on the non-template strand.

All looping reporters (Supplementary Fig. 1) used in the loop assistance experiments were as Priest et al.[13], described briefly below:

The 300 bp–5.6 kb enhancer reporters and enhancer-minus controls used in the enhancer activation experiments (Supplementary Fig. 1) were derived from those of Priest et al.[13], by replacing the PlacUV5 promoter with the E. coli glnAp2 promoter (−36 to +21), and replacing the lacOid operator with an enhancer module comprising the first and the second NtrC sites of glnA (−100 to −151).

All lacZ reporters were integrated into the host chromosome at the λ attB site with chloramphenicol (20 µg mL⁻¹) selection.

The 11.7 kb NtrC activation reporter (Supplementary Fig. 1) was generated in two steps. First, an enhancer-minus pIT-HF-CL_glnAp2_lacZ*(O2−) reporter was integrated into the host chromosome at the λ attB site. Second, the NtrC enhancer was inserted into a kanamycin resistance module flanked by transcription terminators[54] in a pUC57 plasmid, and a derived PCR product was inserted into the gap region between the moaA and ybhK genes (EcoCyc MG1655: 816,719–816,720) by recombineering with kanamycin (20 µg mL⁻¹) selection.

**Sp dCas9 and dCas9_Zip expression constructs.** The catalytically inactive S. pyogenes dCas9 was PCR amplified from pdCas9-bacteria (Addgene plasmid # 44249) and assembled together with a proA promoter[22] into the pIT3-SH integration plasmid (Supplementary Fig. 1). Heterodimerizing Sp dCas9_Zip was constructed by fusing an engineered 43 amino acid (a.a.) long coiled-coil peptide (EE$_{12}$RR$_{345}$L) containing five leucine zipper heptad repeats[15] at the C terminus of Sp dCas9, connected via an 8 a.a. GGGGSGGR linker. To vary the expression levels of Sp dCas9_Zip, the proA promoter was replaced with either the proC or pro1 promoter[22]. Single-copy integrants of pIT3-SH_proA_Sp dCas9, pIT3-SH_proC_Sp dCas9_Zip, pIT3-SH_proA_Sp dCas9_Zip, and pIT3-SH_pro1_Sp dCas9_Zip at the φHK022 attB site were selected with 20 µg mL⁻¹ spectinomycin and used to provide different concentrations of wild-type or heterodimerizing Sp dCas9.

**St dCas9 and dCas9_Zip expression constructs.** The catalytically inactive S. thermophilus CRISPR1 dCas9 was PCR amplified from DS-ST1casN- (Addgene plasmid # 48659) and assembled together with the proA promoter[22] into the pIT4-KT integration plasmid (Supplementary Fig. 1). Heterodimerizing St dCas9_Zip was constructed by fusing an engineered 43 a.a. long coiled-coil peptide (RR$_{12}$EE$_{345}$L)[15] at the C terminus of St dCas9 with an 8 a.a. GGGGSGGR linker. The (EE$_{12}$RR$_{345}$L) and (RR$_{12}$EE$_{345}$L) leucine zipper pair[15] were designed to form very stable heterodimers, while minimizing homodimer formation.

To vary the expression levels of St dCas9_Zip, the proA promoter was replaced with either the proC or pro1 promoter[22]. Single-copy integrants of pIT4-KT_proA_St dCas9, pIT4-KT_proC_St dCas9_Zip, pIT4-KT_proA_St dCas9_Zip, and pIT4-KT_pro1_St dCas9_Zip at the φP21 attB site were selected with 20 µg mL⁻¹ kanamycin and used to provide different concentrations of wild-type or heterodimerizing St dCas9. The kanamycin resistance gene cassette was subsequently removed using Flp recombinase supplied from the pE-FLP plasmid[52].

**Split-GFP constructs.** The pSp dCas9_Zip_gfp_C plasmid (Supplementary Fig. 2) was constructed by three-fragment Gibson isothermal assembly consisting of the proC promoter Sp dCas9_Zip fragment amplified from pIT3-SH_proC_Sp dCas9_Zip (see above), a gBlock fragment (IDT) carrying the C-terminal half (a.a. 158–238) of a folding reporter GFP (frGFP)[23] with a 6 a.a. TSGGSG linker, and a fragment carrying a CloDF13 ori and a spectinomycin resistance gene obtained by PCR amplification from the DS-ST1casN- plasmid.

The complementary pSt dCas9_Zip_gfp_C plasmid (Supplementary Fig. 2) was made by first linearizing the DS-ST1casN- plasmid with FspAI/FseI and assembling with a gBlock (IDT) containing an engineered RR$_{12}$EE$_{345}$L leucine zipper[15] and the N-terminal half (a.a. 1–157) of the frGFP joined via a 6 a.a. TSGGSG linker. This construct was then re-digested with KpnI/FseI to replace the CloDF13 ori and spectinomycin resistance gene cassette with a p15a ori and kanamycin resistance gene cassette amplified from the pUHA-1 plasmid (gift from H. Bujard, Heidelberg University, Germany).

**Sp_St dCas9 fusion expression construct.** The Sp_St dCas9 fusion expression construct pProC_Sp_dCa9s_St_dCas9 (Supplementary Fig. 2) was derived from pSp dCas9_Zip_GFP_C by replacing the C-terminal Zip-GFP_C fusion with St

dCas9, which had been PCR amplified from DS-ST1casN-, using primers designed to join Sp and St dCas by a 15 a.a. (GGGGS)₃ flexible linker[55]. To vary the expression levels of the Sp_St dCas9 fusion, the proC promoter was replaced by either the proA or pro1 promoter[22].

**Dual-sgRNA expression constructs.** The pDual-sgRNA expression construct (Supplementary Fig. 2) was derived from pgRNA-bacteria (Addgene plasmid # 44251). First, a module consisting of an engineered promoter (BioBrick part BBa_J23119) driving expression of a St sgRNA scaffold (the St1m1 variant)[11], followed by synthetic tandem T3 early and T7 early terminators[54], was inserted into the AatII/EcoRI sites of pgRNA-bacteria. The inserted St sgRNA expression module was placed in a divergent orientation relative to a similar Sp sgRNA expression module and was separated from the Sp sgRNA expression module via strong M13 central and λ tR2 terminators[54]. Second, the high copy number ColE1 ori of pgRNA-bacteria plasmid was replaced by the medium copy number p15a ori. To vary the expression levels of the Sp and St sgRNAs, the strong J23119 promoter was replaced by J23114, J23113, or J23112 promoters, respectively. All spacer sequences used in this study are listed in Supplementary Table 1.

**Multiplex sgRNA expression constructs.** The quadruplicate sgRNA expression construct (Supplementary Fig. 2) was derived from pDual-sgRNA by first introducing a unique NotI site between p15a ori and the ampicillin resistance gene cassette. This pDual-sgRNA_NotI plasmid was linearized at NotI, and re-circularized by using Gibson assembly to incorporate a second Sp sgRNA + St sgRNA expression module, amplified from pDual-sgRNA.

**Minimal medium LacZ assays.** Microtiter plate-based LacZ assays were carried out as previously described[24]. Briefly, cultures were grown in microtitre plates in M9MM supplemented with 2 mM MgSO₄, 0.1 mM CaCl₂, 0.01 mM (NH₄)₂Fe (SO₄)₂, and 0.4% glycerol. After reaching mid-log phase, 20 µl of culture was combined with buffer containing (per well) 30 µL of M9MM, 150 µL of TZ8 (100 mM Tris-HCl, pH 8.0, 1 mM MgSO₄, 10 mM KCl), 40 µL of ONPG (o-nitrophenyl-β-D-galactoside 4 mg mL⁻¹ in TZ8), 1.9 µL of 2-mercapoethanol, and 0.95 µL of polymyxin B (20 mg mL⁻¹; Sigma). Assays were performed in triplicate from independent colonies and repeated on at least three different days (n = 9).

**Chromosome conformation capture assays.** The 3C assays were performed as previously described[56] with slight modifications. Briefly, 500 µL of OD$_{600}$ ~0.6 bacterial culture, grown in M9MM, was cross-linked with formaldehyde (1% final v/v) for 30 min at room temperature, followed by 30 min on ice, and then quenched with glycine (125 mM final). Cells were collected by centrifugation, washed twice with 1 mL M9MM, and lysed with 30 µL 25 mg mL⁻¹ lysozyme (Sigma-Aldrich, L6876) at room temperature for 10 min. Ten percent SDS was subsequently added to a final concentration of 0.5% and incubated with the lysed cells for further 10 min. Five microliters of this cell lysate were then added to 47 µL of digestion mix (5.3 µL 10x New England Biolab (NEB) Buffer 2, 5.3 µL 10% Triton X-100, and 35 µL MQ water), and incubated at room temperature for further 10 min before 2.4 µL of BsaWI (NEB, 10 units per µL) was added. Cell lysate was digested at 60 °C for 1 h. Following digestion, 10% SDS was added to a final concentration of 0.9% and the solution incubated at room temperature for 10 min. The whole digestion reaction was then added to a ligation mix consisting of 74.5 µL 10% Triton X-100, 0.13 µL 10 mg mL⁻¹ bovine serum albumin, 5.3 µL 10 mM ATP, 76.3 µL 10x T4 DNA ligase buffer, 603 µL MQ water, and 1 µL of 2000 units per µL NEB T4 DNA ligase. Ligation was carried out at 16 °C for 24 h. Following ligation, 0.5 M EDTA was added to a final concentration of 10 mM. Cross-links were reversed via the addition of 2.5 µL of 20 mg mL⁻¹ Proteinase K (Sigma-Aldrich, P6556) and an overnight incubation at 16 °C. DNA was purified via two rounds of phenol-chloroform-isoamyl alcohol extraction (Sigma-Aldrich, P2069) and a single round of ethanol precipitation. The typical yield was 50 ng µL⁻¹ in 50 µL of MQ water. The relative contact frequencies between the anchor fragment and various BsaWI restriction digest fragments were assessed by qPCR. A DNA fragment from the rho gene served as a loading control. All experiments were performed in technical triplicates for each of three biological replicates. The primer sequences can be found in Supplementary Table 2.

**RNA isolation and reverse transcription.** Bacterial cells were grown in M9MM to late exponential growth phase cultures (OD$_{600}$ ~0.8). An aliquot of 800 µL of bacterial culture was then stabilized with RNAprotect Bacteria Reagent (QIAGEN) and total RNA extracted using RNeasy Mini Kit (QIAGEN) according to the manufacturer's instructions. Traces of contaminating DNA were removed from isolated RNA using the Turbo DNA-free kit (Ambion) as per the manufacturer's protocol. RNA concentration and purity, determined by 260/280 and 260/230 ratios, were measured with a NanoDrop 2000 spectrophotometer. An aliquot of 1 µg of DNA-free RNA was then reverse-transcribed to cDNA using QuantiTect Reverse Transcription kit (QIAGEN) according to the manufacturer's instructions. The primer sequences as well as primer efficiencies and amplicon sizes are listed below. The E. coli gyrA gene was used as a reference gene as its expression did not change across different test strains. All experiments were performed in technical

triplicates for each of three biological replicates. Details of primers used in qPCR experiments are given in Supplementary Table 3.

**Statistics**. For real-time qRT-PCR experiments, statistical differences were evaluated by a paired one-tailed Student's $t$ test on the $\log_{10}$ transformed data, with the level of significance set at $p < 0.05$.

**Data availability**. The data that support the findings of this study are available from the corresponding author on reasonable request.

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

## Acknowledgments

We are grateful to members of the Shearwin lab, particularly David Priest, for discussions. This work was supported by the National Health and Medical Research Council via a project Grant (GNT1100653) to I.B.D., K.E.S., and N.H. and the Australian Research Council via a Discovery Early Career Researcher Award to N.H. (DE150100091) and Discovery Grant to K.E.S. (DP160101450).

## Author contributions

N.H., K.E.S., and I.B.D. designed the experiments. N.H. performed the experiments. N.H., K.E.S., and I.B.D. analyzed the experiments. I.B.D. did the modeling. N.H., K.E.S., and I.B.D. wrote the paper.

## Additional information

**Competing interests:** The authors declare no competing financial interests.

