## [Peer Review File · Nature Communications]

Reviewers' comments:

Reviewer #1 (Remarks to the Author):

In this manuscript Hao et al describe a method to investigate DNA looping. The authors exploit dimerization of Cas9 orthologues to induce DNA looping in bacteria. The experimental evaluation of the method is accompanied by a modelling analysis.

This study reports on a novel molecular approach to investigate DNA looping an event which is relevant in cis acting control of transcriptional events. Indeed, these mechanisms are extremely relevant within the context of 3D nuclear control of gene function yet an accurate in vivo investigation tool is still missing and this is a timely study within this context. The study is well performed and the method proposed by Hao et al would in principle provide a powerful tool for the discovery of cis and trans acting elements in the genome.

Proving that the method can be applied in eukaryotic cells would dramatically raise the impact of this work and would elevate the manuscript sufficiently for publication in this journal.

Major Critiques:

1) Does the "programmable DNA looping" work in eukaryotic systems?

2) Validation of the DNA looping experimental system was performed in bacteria using a reporter gene (lac Z). The use of endogenous bacterial genomic loci rather than reporter genes (even though chromosomally integrated) is necessary to evaluate the real applicability of the model.

3) Several sections of the manuscript are cryptic and hard to understand to non-experts. Figure S2 is barely explained, the roadblocking assay (line 147) even if referenced requires a minimal description. Figure 1A is hard to interpret for a wider readership, it would better moved to supplementary section with detailed description even for non-expert.

Minor critiques:

1) Line 93 "no looping improvement is gained by having different values of KDNA... ": this sentence is in apparently in contrast with the rest of the study (for example see line 125)

2) The DNA interaction emerges as one of the weakness of the model. Indeed, the authors speculate that based on former studies (refs 38 and 39) Cas9 mutations with hydrophilic amino acids may strengthen overall DNA binding thus potentially improving the DNA looping model. In ref 38 (Slaymaker et al.) such variants are reported and testable in the DNA looping system here proposed.

3) The model is based on KDNA from Cas9. However the Cas9 used are modified with leucine zipper domain. Is the KDNA preserved?

4) Discussion section relative to potential applications of bivalent Cas9 in eukaryotic cells is quite limited and does not place the DNA loop experimental set up in the proper context such as in vivo analysis of cis/trans acting genetic elements in nuclear functions.

5) line 157: Fig 2D instead of 1D

Reviewer #2 (Remarks to the Author):

In this paper, Hao et al. is applying catalytically inactive dCas9s from orthogonal CRISPR systems

to achieve directed DNA loop formation. Long-range interactions between regulatory DNA elements are critical determinants of gene regulation. Thus, an ability to manipulate DNA loop formation and thus achieving chromatin engineering is important and any efforts towards this goal are significant. To this end, the design of this study is highly novel and interesting. The manuscript is well written and I enjoyed reading it. Hao et al have also recently reported (NAR, 2017) where did mathematical modeling of DNA loop formations. In this manuscript, they have also included several equations and modeling. I think this manuscript should only focus on dCas9-mediated chromatin looping and show whether this approach is working or not. Here are some minor concerns that I have.

- I think the modeling part is not adding much to the manuscript and is not relevant to the main idea of this manuscript, which is, as the title suggests, "Programmable DNA looping using engineered bivalent dCas9 complexes."
- The model-based calculations and experimental data are presented concurrently and make it difficult to follow what has actually been done experimentally.
- The data and Fig 2C and Fig 2E seem contradictory. 2C shows that High Promoter is blocking the read through much more than intermediate or low activity promoter. But 2E shows that promoter strength is not altering read through?
- The LacZ Units values should be written in figures such as 3b, 4c, 5d. It is not clear whether the value axis is linear or log scaled and in each case only the top value is indicated.
- Why is Fig 5b shown? Is this an experimental data or modeling data? Why is it important to show in this context?

Reviewer #3 (Remarks to the Author):

The authors describe a new experimental approach allowing formation of DNA loops between every two desired DNA regions in the bacterial genome using cleavage-defective Cas9 proteins of different sequence specificity that either can heterodimerize or are fused together in one bivalent protein. This approach allows formation of the loops de novo or stabilization of existing protein-mediated DNA loops of various sizes in the range up to over 10 kb in vivo. This is an important topic because the majority of gene regulation in higher eukaryotes occurs through distant interactions involving looping of intervening DNA/chromatin, and the ability to direct the looping could result in development of new important regulatory eukaryotic networks.

This high-quality, well-controlled work is addressing an important issue and the presented data fully support the conclusions. The developed tools are important and novel, and the manuscript is well written. However, the work is rather technical and does not describe any new biological process or mechanism. My other primary concern is that the loop formation in vivo was established using rather indirect assays. Below I elaborate on these and some other concerns:

1. In the experimental part of the work, the authors either monitor binding of Cas9 proteins to DNA (by monitoring the yield of lacZ) and the effect of Cas9 proteins on lac repressor-dependent gene regulation. Neither of the assays directly monitors loop formation in vivo. The authors should confirm loop formation using a more direct assay (e.g. a 3C-type method).
2. Abstract: "Although DNA looping can now be efficiently detected, tools to readily manipulate DNA looping are lacking." This sentence should be corrected. Tools to manipulate DNA looping have many limitations, but they do exist and the authors used some of them (LacR looping).
3. Figs 3 & 4: The observed effects are rather small. The effects would be more robust if loop assistance over larger distances was tested.
4. The work is rather technical and does not describe any new biological process or mechanism.

Reviewer #1 (Remarks to the Author):

In this manuscript Hao et al describe a method to investigate DNA looping. The authors exploit dimerization of Cas9 orthologues to induce DNA looping in bacteria. The experimental evaluation of the method is accompanied by a modelling analysis. This study reports on a novel molecular approach to investigate DNA looping, an event which is relevant in cis acting control of transcriptional events. Indeed, these mechanisms are extremely relevant within the context of 3D nuclear control of gene function yet an accurate in vivo investigation tool is still missing and this is a timely study within this context. The study is well performed and the method proposed by Hao et al would in principle provide a powerful tool for the discovery of cis and trans acting elements in the genome. Proving that the method can be applied in eukaryotic cells would dramatically raise the impact of this work and would elevate the manuscript sufficiently for publication in this journal.

Major Critiques:

1) Does the “programmable DNA looping” work in eukaryotic systems?

While our manuscript was under review, a Nature Communications paper by Morgan et al. (DOI: 10.1038/ncomms15993) has demonstrated Cas9 looping in human cell lines, using a different dimerization domain from the one we have used.

2) Validation of the DNA looping experimental system was performed in bacteria using a reporter gene (lac Z). The use of endogenous bacterial genomic loci rather than reporter genes (even though chromosomally integrated) is necessary to evaluate the real applicability of the model.

We have now performed an experiment to show DNA looping regulation of endogenous *E. coli* genes by the dimeric Cas9 reagent (Fig 7 of the revised manuscript).

3) Several sections of the manuscript are cryptic and hard to understand to non-experts. Figure S2 is barely explained, the roadblocking assay (line 147) even if referenced requires a minimal description. Figure 1A is hard to interpret for a wider readership, it would better moved to supplementary section with detailed description even for non-expert.

We have revised the manuscript to include more description of the transcriptional roadblocking assay.

We think the modeling of DNA looping by the bivalent Cas reagents is an essential part of the paper for readers who wish to understand the system quantitatively. The modeling allows the establishment of the key determinants for maximal looping efficiency and provides a useful guide for the further optimization of the Cas looping reagents. Thus, we are keen to keep Fig 1A as part of the main figure. However, we have revised the manuscript to make this part more clear.

Minor critiques:

1) Line 93 “no looping improvement is gained by having different values of KDNA... “: this sentence is in apparently in contrast with the rest of the study (for example see line 125)

This sentence was confusing. We have changed it to: “In our model, we made the simplifying assumption that K_{DNA} , θ , and $[C_{tot}]$ are the same for both *Sp* and *St* dCas9.”

2) The DNA interaction emerges as one of the weakness of the model. Indeed, the authors speculate that based on former studies (refs 38 and 39) Cas9 mutations with hydrophilic amino acids may strengthen overall DNA binding thus potentially improving the DNA looping model. In ref 38 (Slaymaker et al.) such variants are reported and testable in the DNA looping system here proposed.

In ref 38, Slaymaker et al. reported the creation of *Sp*Cas9 S845K and L847R mutations, which strengthen the interactions between Cas9 and the non-target strand of DNA. As expected, these mutations increased off-target cleavage at the EMX1(1) site. It would certainly be interesting to see what effect these mutations have on bivalent Cas reagents in terms of their ability to loop DNA. However, we see this as a question for future experiments.

This has now been incorporated into the Discussion.

3) The model is based on KDNA from Cas9. However the Cas9 used are modified with leucine zipper domain. Is the KDNA preserved?

It is true that the K_{DNA} value used in the model is based on estimations for unmodified Cas9. As the leucine zipper domains do not contribute to DNA binding, this value is likely to represent an upper limit for the dCas9-DNA dissociation constant. However, based on the results of the transcriptional roadblock assay, it is expected that the K_{DNA} values for dCas9 and for the dCas9 leucine zipper fusions are very similar. In addition, the Cas looping model predicts that DNA looping by bivalent dCas9 is ~40% at 1.4kb and ~14% at 4.7kb, which are consistent with those estimated from the loop assistance experiments (~40% and 17% respectively). Note that this experimental estimation of looping does not involve an estimate of K_{DNA} .

4) Discussion section relative to potential applications of bivalent Cas9 in eukaryotic cells is quite limited and does not place the DNA loop experimental set up in the proper context such as in vivo analysis of cis/trans acting genetic elements in nuclear functions.

We believe that these points are well discussed in the recent paper by Morgan et al. (DOI: 10.1038/ncomms15993), and do not need to be repeated here. Instead, we have raised the additional possibilities of bivalent Cas reagents being used to improve specificity of Cas action or to join separate DNA molecules.

5) line 157: Fig 2D instead of 1D

This has been corrected.

Reviewer #2 (Remarks to the Author):

In this paper, Hao et al. is applying catalytically inactive dCas9s from orthogonal CRISPR systems to achieve directed DNA loop formation. Long-range interactions between regulatory DNA elements are critical determinants of gene regulation. Thus, an ability to manipulate DNA loop formation and thus achieving chromatin

engineering is important and any efforts towards this goal are significant. To this end, the design of this study is highly novel and interesting. The manuscript is well written and I enjoyed reading it. Hao et al have also recently reported (NAR, 2017) where they did mathematical modeling of DNA loop formations. In this manuscript, they have also included several equations and modeling. I think this manuscript should only focus on dCas9-mediated chromatin looping and show whether this approach is working or not. Here are some minor concerns that I have.

- I think the modeling part is not adding much to the manuscript and is not relevant to the main idea of this manuscript, which is, as the title suggests, “Programmable DNA looping using engineered bivalent dCas9 complexes.

We think the modeling of DNA looping by the bivalent Cas reagents is an essential part of the paper for readers who wish to understand the system quantitatively. The modeling allows the establishment of the key determinants for maximal looping efficiency and provides a useful guide for the further optimization of the Cas looping reagents.

- The model-based calculations and experimental data are presented concurrently and make it difficult to follow what has actually been done experimentally.

We have revised the manuscript in an attempt to clearly separate the modeling based calculation and the experimental data.

- The data and Fig 2C and Fig 2E seem contradictory. 2C shows that High Promoter is blocking the read through much more than intermediate or low activity promoter. But 2E shows that promoter strength is not altering read through?

The data of Fig 2C and Fig 2E are not contradictory. As the reviewer has noted, figure 2C shows that the ability of dCas9 to act as a transcriptional roadblock depends on its concentration, the stronger the promoter that drives dCas9 expression (i.e. the higher the dCas9 concentration), the stronger the roadblock effect. In contrast, the promoter strength on the x-axis in Fig 2E refers to the promoter that drives sgRNA expression. This data shows that in our system, the dCas9 is fully saturated with its cognate sgRNAs. Hence, roadblocking by dCas9 is not changed when the promoter expressing the sgRNA is weakened ~2500-fold.

- The LacZ Units values should be written in figures such as 3b, 4c, 5d. It is not clear whether the value axis is linear or log scaled and in each case only the top value is indicated.

The LacZ unit axes figures 3b, 4c, and 3d are linear. We have now amended these figures (now Figures 4B, 5C and 4D), and added the interval LacZ unit values where appropriate.

- Why is Fig 5b shown? Is this an experimental data or modeling data? Why is it important to show in this context?

The data presented in Figure 5B (now 6B) are experimental data (n=9, error bar =95% CI). We have changed the presentation of the figure in an attempt to more clearly indicate the error bars.

This figure establishes that the activity of the NtrC-bound enhancer at the σ^{54} promoter is dependent on the distance between them. The decreasing activity with increasing separation reflects the decreasing probability of DNA looping. Having

demonstrated this, we subsequently show that enhancer-mediated reporter gene activation can be significantly increased when the enhancer site is brought closer to the promoter using bivalent dCas9 looping reagents (Fig. 6CD and Fig 7C).

Reviewer #3 (Remarks to the Author):

The authors describe a new experimental approach allowing formation of DNA loops between every two desired DNA regions in the bacterial genome using cleavage-defective Cas9 proteins of different sequence specificity that either can heterodimerize or are fused together in one bivalent protein. This approach allows formation of the loops de novo or stabilization of existing protein-mediated DNA loops of various sizes in the range up to over 10 kb in vivo. This is an important topic because the majority of gene regulation in higher eukaryotes occurs through distant interactions involving looping of intervening DNA/chromatin, and the ability to direct the looping could result in development of new important regulatory eukaryotic networks.

This high-quality, well-controlled work is addressing an important issue and the presented data fully support the conclusions. The developed tools are important and novel, and the manuscript is well written. However, the work is rather technical and does not describe any new biological process or mechanism. My other primary concern is that the loop formation in vivo was established using rather indirect assays. Below I elaborate on these and some other concerns:

1. In the experimental part of the work, the authors either monitor binding of Cas9 proteins to DNA (by monitoring the yield of lacZ) and the effect of Cas9 proteins on lac repressor-dependent gene regulation. Neither of the assays directly monitors loop formation in vivo. The authors should confirm loop formation using a more direct assay (e.g. a 3C-type method).

We have now performed the 3C experiment as requested (new figure 3), directly confirming Cas-mediated looping.

2. Abstract: "Although DNA looping can now be efficiently detected, tools to readily manipulate DNA looping are lacking." This sentence should be corrected. Tools to manipulate DNA looping have many limitations, but they do exist and the authors used some of them (LacR looping).

Corrected

3. Figs 3 & 4: The observed effects are rather small. The effects would be more robust if loop assistance over larger distances was tested.

As acknowledged in the first section of the Discussion, looping by the bivalent Cas reagents is weaker than well-characterized looping proteins such as *E. coli* Lac repressor and lambda CI repressor at a similar looping distance (Priest et al. 2014 DOI:10.1073/pnas.1410764111). However, Cas looping can be easily improved by multiplexing, has the important advantage of programmability, allowing it to be applied to existing DNA sequences, and should be readily portable to other cell types.

Furthermore, in this study, we also attempt to use modelling-assisted experimental design to establish the key determinants for maximal looping efficiency by bivalent Cas reagents and suggest that one way to increase the looping efficiency is to

increase the binding affinity between dCas9 and DNA (together with low dCas9 concentration).

Our previous measurements with LacI and CI indicate that the fraction of looping falls roughly proportionally with distance (Priest et al. 2014 DOI: 10.1073/pnas.1317817111). Thus, we would expect roughly 5% looping for a single bivalent Cas at 10 kb. The regulatory impact of the Cas looping is modest in the LacI repression assay, in part because Cas looping is not strong and in part because the LacI loop between *O2* and *Oid* forms readily in the absence of any Cas looping. Even if the Cas loop formed with 100% efficiency, we expect only a ~50% reduction in promoter activity for the 1.4 kb spacing. In the bacterial enhancer activation experiments, where the bivalent Cas aids activation of the promoter, the regulatory effect of the Cas looping is larger (a ~2-fold effect for single loops and a ~4-fold effect for double loops) and works at a distance of 12 kb.

4. The work is rather technical and does not describe any new biological process or mechanism.

The primary goal of this study was to provide a proof-of-principle that the bivalent Cas reagents could be used to loop DNA *in vivo*, and to provide a quantitative analysis of such looping, to provide a useful tool for a broad scientific community to study DNA looping.

While not setting out to find new biological processes, the study does provide the first demonstration that *de novo* DNA looping by bivalent Cas looping reagents can improve activation by a bacterial enhancer located far from the promoter.

We believe that this study amply meets the aims and scope of *Nature Communications*, that “an Article is a novel and important research study of high quality and of interest to that specific research community.” As this reviewer states: “This high-quality, well-controlled work is addressing an important issue and the presented data fully support the conclusions. The developed tools are important and novel,...”.

REVIEWERS' COMMENTS:

Reviewer #1 (Remarks to the Author):

The authors have integrated new experimental work that properly addressed the issues raised during the initial revision. The data recently published on DNA looping may compromise the novelty of this work. Nevertheless, the value of this study in a bacterial set up is preserved. The work is now suitable for publication.

Reviewer #3 (Remarks to the Author):

The authors have successfully replied to all comments, conducted the requested experiments, and considerably improved the manuscript. The new data are consistent with the ones presented in the original manuscript.

Reviewer #1 (Remarks to the Author):

The authors have integrated new experimental work that properly addressed the issues raised during the initial revision. The data recently published on DNA looping may compromise the novelty of this work. Nevertheless, the value of this study in a bacterial set up is preserved. The work is now suitable for publication.

Reviewer #3 (Remarks to the Author):

The authors have successfully replied to all comments, conducted the requested experiments, and considerably improved the manuscript. The new data are consistent with the ones presented in the original manuscript.

Both reviewers are satisfied that the new experiments have successfully addressed their initial concerns and recommend the publication of this study in *Nature Communications*.